# Ten-Year Outcome and Development of Virtual-Assisted Lung Mapping in Thoracic Surgery

**DOI:** 10.3390/cancers15071971

**Published:** 2023-03-25

**Authors:** Masaaki Nagano, Masaaki Sato

**Affiliations:** Department of Thoracic Surgery, The University of Tokyo Graduate School of Medicine, 7-3-1 Hongo, Bunkyo-ku, Tokyo 113-8655, Japan

**Keywords:** virtual-assisted lung mapping (VAL-MAP), sublobar resection, electromagnetic navigation, indocyanine green, microcoil

## Abstract

**Simple Summary:**

Virtual-assisted lung mapping (VAL-MAP) is a preoperative bronchoscopic multispot dye-marking technique that has demonstrated efficacy and good outcomes over the last 10 years. However, conventional VAL-MAP has certain limitations. Several new techniques have been developed to address the problems associated with conventional VAL-MAP and have shown good results. Here, we summarize not only the history and outcomes of VAL-MAP, but also the latest update of this preoperative marking technique using bronchoscopy.

**Abstract:**

Virtual-assisted lung mapping (VAL-MAP) is a preoperative bronchoscopic multispot dye-marking technique used in sublobar lung resection of barely palpable lung nodules. This review summarizes the history and outcomes of the VAL-MAP procedure. VAL-MAP was developed in 2012, and long-term outcomes of lung resection using VAL-MAP have recently been verified. Problems associated with conventional VAL-MAP include a prerequisite of post-mapping computed tomography (CT), occasional inability to see dye marks during surgery, and infrequent resection failure due to deep resection margins; various techniques have been developed to address these issues. VAL-MAP using electromagnetic navigation bronchoscopy with on-site adjustment can omit post-mapping CT. The use of indocyanine green in VAL-MAP has increased the success rate of marking detection during surgery without causing additional complications. VAL-MAP 2.0—a three-dimensional mapping technique that involves the intrabronchial placement of a microcoil—has increased the accuracy of sublobar resection, particularly for deeply located tumors. Although these promising new techniques have some limitations, they are beneficial for sublobar lung resection.

## 1. Introduction

Lung cancer is a leading cause of cancer-related death, accounting for nearly 2.2 million cases in 2020 [1]. The development of several chemotherapies and molecular targeted therapies has improved the prognosis and progression-free survival of patients with advanced lung cancer [2,3,4]. However, the ultimate goals for thoracic surgeons should be to increase the likelihood of early detection and optimal surgical management of lung cancer, eventually improving the survival in patients with lung cancer. The recent widespread use of computed tomography (CT) has enabled for the detection of small pulmonary nodules [5]. The National Lung Screening Trial (NLST) was a randomized controlled trial between low-dose CT and single-view posteroanterior chest radiography to determine whether screening with low-dose CT could reduce mortality from lung cancer. This trial showed 247 deaths from lung cancer per 100,000 person years in the low-dose CT group compared with 309 deaths per 100,000 person years in the radiography group, demonstrating that three annual screenings with low-dose CT could reduce mortality from lung cancer by 6.7%. Furthermore, two recent large clinical studies have shown that sublobar resection has an equivalent or higher survival rate in small lung cancers than lobectomy [6,7]. The JCOG0802/WJOG4607L trial was a multicenter, open-label, randomized, controlled trial at 70 institutions in Japan to compare the results of lobectomy and segmentectomy for clinical stage IA non-small cell lung cancer (tumor diameter ≤2 cm; consolidation-to-tumor ratio >0.5). This study showed that the 5-year overall survival was 94.3% for segmentectomy and 91.1% for lobectomy and that superiority and non-inferiority in overall survival were confirmed using a stratified Cox regression model (HR 0.663; 95% CI 0.474–0.927 [6]. Cancer and Leukemia Group B (CALGB) 140,503 was a multicenter, randomized trial at 83 institutions in the United States, Canada, and Australia to compare the results of lobectomy and sublobar resection, including wedge resection and segmentectomy for clinical stage IA non-small cell lung cancer with a tumor size of 2 cm or less. This study showed that sublobar resection was noninferior to lobar resection for disease-free survival (hazard ratio for disease recurrence or death, 1.01) and that the 5-year overall survival was 80.3% after sublobar resection and 78.9% after lobar resection, indicating the non-inferiority of sublobar resection to lobectomy with respect to disease-free survival and overall survival [7].

While the importance of sublobar resection for small lung cancer, which is often detected via a low-dose CT scan, it has been notified recently that local recurrence has been a significant issue to overcome. The JCOG0802/WJCOG4607L trial showed that significantly more locoregional relapses occurred in patients who had segmentectomy (11%) than in those who had lobectomy (5%). The CALGB 140,503 trial also showed that locoregional recurrence was significantly higher in sublobar resections (13.4%) than in lobectomies (10.0%). Furthermore, the development of minimally invasive surgeries, such as video-assisted thoracic surgery and robot-assisted thoracic surgery, has made it more challenging to palpate and detect small pulmonary nodules intraoperatively [8,9]. These results indicate that an effective preoperative localization technique is essential for confirming the adequacy of the surgical margin and preventing local recurrence.

Virtual-assisted lung mapping (VAL-MAP) is a preoperative bronchoscopic multispot dye-marking technique used to identify barely identifiable pulmonary nodules and draw appropriate resection lines in sublobar lung resection [10]. This unique technique was developed in 2012 and has shown efficacy and good outcomes over the last 10 years. However, the problems that must be solved have also become clear: a prerequisite of post-mapping CT, occasional inability to see dye marks during surgery, and infrequent resection failure due to deep resection margins. Several new techniques have been developed to address these problems and have shown good results. Here, we briefly describe the latest update in preoperative marking techniques using bronchoscopy.

## 2. Details of Conventional VAL-MAP (VAL-MAP 1.0)

The steps of VAL-MAP 1.0 are summarized in Figure 1 [11]. VAL-MAP consists of planning, bronchoscopic dye injection, post-mapping CT scans, and operation. First, a three-dimensional reconstruction of CT data and virtual bronchoscopic images are created using high-resolution CT scans. Lung mapping is planned by surgeons for the purpose of identifying tumors and indicating the resection lines in sublobar lung resection. How many marks should be performed depends on the purpose and characteristics of operation. Workstations and software by which we generate virtual bronchoscopy and three-dimensional images are ZiostationVR (Ziosoft, Tokyo, Japan), Zioterm 2009VR (Ziosoft), and Synapse VincentVR (Fujifilm Medical, Tokyo, Japan). The patients then undergo a bronchoscopic procedure under mild sedation and local anesthesia on the day of surgery or 1 day before surgery. After bronchoscopy is advanced toward the target bronchi using virtual images as a guide, a metal-tipped catheter (PW-6C-1 or PW-6P-1; Olympus, Tokyo, Japan) preloaded with 1 mL (4 mg) of Indigo carmine (Daiichi-Sankyo, Tokyo, Japan) is inserted through the working channel of the bronchoscope. The tip of the catheter is confirmed to have reached the visceral pleura, and then indigo carmine is gently injected into the target bronchus via the catheter. This procedure is repeated several times as planned. Post-mapping CT images are acquired and developed into three-dimensional images for surgery to confirm the actual locations of the marks and tumor. Finally, the dye marks can be readily recognized as blue spots on the surface of the lung during surgery.

Virtual-assisted lung mapping is designed using the three-dimensional reconstruction of computed tomography (CT) data and virtual bronchoscopic images. Bronchoscopic dye injection is performed on the day of surgery or 1 day before surgery via a metal-tipped catheter preloaded with 1 mL of indigo carmine (black arrow). Post-mapping CT images are taken and developed into three-dimensional images for surgery to confirm the actual locations of the marks (arrowhead) and the tumor. Finally, the operation is conducted with the help of dye marks, which can be readily recognized as blue spots on the surface of the lung (white arrow). This figure is reprinted from the reference [11].

## 3. Comparison of CT-Guided Localization and Bronchoscopic Marking

There are many different localization methods for hardly palpable lung nodules, and two major image-guided localization techniques are CT-guided localization and bronchoscopic marking. CT-guided localization techniques are widely used for preoperative markings with various localized materials, including hook-wires, microcoils, fiducial markers, dyes, and radiotracers. The advantages and disadvantages of each material are shown in Table 1. Hook-wire is the most common material used in CT-guided localization [12,13,14,15]. The success rate of this method is 92.5–97.6%, and surgeons can visually detect the localized site intraoperatively without any radiation exposure. Dislodging of the hook-wire is one of the major drawbacks, with a rate of 2.4–7.5%, and the occurrence of pneumothorax (7.5–40%) and lung hemorrhage (10.3–36%) are also relatively high [12,13,14,15]. Microcoils [16,17] and fiducial markers [18,19] do not need a wire protruding from the skin after localization, and thus, patients tend not to feel discomfort after the procedure, compared with hook-wire localization. In addition, the inert nature of these metallic materials makes it possible to perform surgery at a different date, providing flexibility between the marking and operating schedules. The major disadvantage of this method is fluoroscopic guidance during the operation, increasing radiation exposure for surgeons. Another disadvantage is that these materials are costly. Dye marking techniques can be easily performed in most institutes without additional costs [20,21]. Moreover, surgeons are not exposed to radiation intraoperatively. However, immediate surgery after CT-guided dye marking may be required because dyes can diffuse easily, leading to incorrect identification of tumors. Anaphylaxis to the dye should be considered, although it is rare [22]. Radiotracer-guided localization is useful because radiotracers are stable for up to 24 h and permit flexibility in scheduling surgery [23,24]. However, facilities where this method is available may be limited because some specific equipment, including radiotracers and gamma probes, are required. Possible radiation exposure during the operation is another disadvantage.

The advantages and disadvantages of bronchoscopic marking techniques are also shown in Table 1. Yanagiya et al. performed a meta-analysis of preoperative bronchoscopic marking for pulmonary nodules [25]. This study included 25 eligible studies from the PubMed and Cochrane Library databases evaluating preoperative bronchoscopic marking for pulmonary resection. Fifteen studies conducted dye marking under electromagnetic navigation bronchoscopy (ENB), four used VAL-MAP, and seven used other marking methods. The overall pooled successful marking rate and successful resection rate were 97% and 98%, whereas the overall pooled rates of pleural injury and pulmonary hemorrhage were 2% and 0%, respectively. These complication rates of bronchoscopic marking are lower than those of CT-guided localization. Moreover, bronchoscopic marking has never caused fatal complications, while CT-guided localization is associated with fatal complications, such as air embolism. ENB is a real-time navigation technique that guides a steerable endoscopic probe to the peripheral lung areas using three-dimensional-reconstructed CT data and sensor location technology [26,27]. An advantage of dye marking under ENB is the high success rate of markings with low rates of complications. However, if the marking fails, localizing and detecting small lung nodules would be extremely difficult. Furthermore, ENB is costly because it requires expensive equipment such as a working channel and guided sensors.

Unlike dye marking under ENB, VAL-MAP 1.0 makes multiple markings. Multiple redundant marks allow surgeons to localize tumors easily and to draw resection lines with sufficient margins, increasing the possibility of successful resection [10]. Another advantage of this method is that it can be performed at a low cost because it only requires reusable catheters and imaging software [25]. In addition, a single-institute retrospective study comparing CT-guided percutaneous needle marking (n = 56) and VAL-MAP 1.0 (n = 97) showed that VAL-MAP 1.0 tended to achieve a higher resection success rate than CT-guided percutaneous needle marking (91.8% vs. 80.4%; *p* = 0.071) [28]. This study also demonstrated that the presence or absence of intraoperative adhesion did not affect the successful resection rate using VAL-MAP 1.0 (85.7% vs. 93.4%; *p* = 0.491), while the successful resection rate using CT-guided percutaneous needle marking was significantly lower in cases with intraoperative adhesion than in cases without intraoperative adhesion (54.5% vs. 86.7%; *p* = 0.048). These results suggest that VAL-MAP 1.0 is more favorable than other localization techniques, despite some disadvantages as listed below.

## 4. Multicenter Studies of VAL-MAP 1.0

Two prospective large-scale multicenter studies demonstrated the sufficient safety of VAL-MAP as well as its efficacy and reproducibility [11,29]. One of them was conducted in 17 centers and enrolled 500 patients who underwent VAL-MAP [11]. This study showed that four patients (0.8%) had major complications associated with VAL-MAP, requiring additional medical management, including fever necessitating the postponement of surgery, pneumonia, and temporary exacerbation of pre-existing cerebral ischemia. Precisely, 1780 markings were designed for VAL-MAP and 1773 markings were actually performed (3.6 ± 1.2 marks per patient). Among them, 1613 marks (91%) were identifiable during operation, and the successful resection rate was approximately 99%. The contribution of VAL-MAP to surgical success was highly evaluated by surgeons when they resected pure ground glass nodules (*p* < 0.0001), tumors less than 5 mm (*p* = 0.0016), and performed complex segmentectomy or wedge resection (*p* = 0.0072). The other prospective study was conducted in 19 registered centers to examine the efficacy of VAL-MAP for obtaining sufficient surgical margins in sublobar lung resection [29]. Two hundred three lesions in one hundred fifty-three patients were resected with the help of VAL-MAP. Successful resection, which was defined as resection of the lesion with margins greater than the lesion diameter or 2 cm without additional resection, was obtained in 178 lesions (87.8%), and the most significant factor affecting successful resection was the depth of the resection margin (*p* = 0.0072).

Moreover, a multicenter retrospective study of long-term results after sublobar resection using VAL-MAP 1.0 showed that the 5-year local recurrence-free and overall survival rates among 264 patients with primary lung cancer were 98.4% and 94.5%, respectively [30]. This study also examined 102 cases with metastatic lung tumors, demonstrating that the 5-year local recurrence-free rates were 94.9% for colorectal cancer, 100% for renal cell cancer, and 90.8% for others, and that the 5-year overall survival rates were 82.3% for colorectal cancer, 100% for renal cell cancer, and 77.3% for others. These results indicate that VAL-MAP 1.0 is associated with reasonable long-term outcomes.

## 5. Problems of VAL-MAP 1.0

Several problems with VAL-MAP 1.0 have been reported (Table 1). First, post-mapping CT is mandatory in VAL-MAP 1.0, because actual marks are likely to deviate from the planned location. A single-center retrospective study examined the accuracy of marking locations predicted using virtual bronchoscopy and elucidated the role of post-mapping CT [31]. The average distance between the predicted and actual marking locations was 3 cm, and the difference was significantly greater in the upper lobes (37.1 ± 20.1 mm) compared to the lower lobes (23.0 ± 6.8 mm). However, this study also showed that all targeted lesions were successfully resected using three-dimensional image guidance based on post-mapping CT, despite the discrepancy between the predicted and actual marking locations. Thus, three-dimensional images created from post-mapping CT data are essential for the successful resection of target lesions. Second, approximately 10% of the marks are invisible during surgery because of patient or technical factors. Heavy smoking history, emphysematous lungs, and severe anthracosis are patient factors associated with the invisibility of VAL-MAP markings [29,32]. Regarding technical issues, a “central injection” (i.e., injection of dye with the catheter tip away from the pleura) has been reported as the primary cause of marking failure [33]. In fact, the muti-center prospective study reported that the main reason for invisible or faint marking was a central injection (24 marks out of 559 marks), followed by severe anthracosis (15 marks), an emphysematous lung (12 marks), and thick pleura (7 marks) [29]. In addition, a recent report has shown that markings in the upper lobe were associated with a significantly increased risk of invisible marks [34]. This report analyzed 857 VAL-MAP markings and assessed the risk factors for invisible markings using multiple logistic regression analysis, showing the significant increase in the risk of obtaining invisible marks for the upper lobe (odds ratio 2.36, 95% confidence interval 1.27−4.39; *p* = 0.0067) and the highest rate of invisible marks in the left B1 + 2c subsegment. Another study compared the successful rate of markings between a supine position (21 patients) and lateral position (27 patients), demonstrating that the rate of visible marks was significantly greater when the markings were performed in the lateral position than in the supine position (88% vs. 57%, *p* = 0.02) [35]. This study also showed that the successful rate of marking tended to be greater in the lateral position than in the supine position, especially among the markings in the dorsal and ventral segments (89% vs. 59%, *p* = 0.06). Third, VAL-MAP 1.0 has an increased risk of insufficient margins as the required depth of the resection line increases. The muti-center prospective study mentioned above revealed that the most significant factor leading to resection failure was the depth of the required resection margin [29]. The required resection depth was calculated as the depth (distance from the closest pleura) + {[diameter × 2 (tumor < 2 cm)] or [diameter + 2 cm (tumor > 2 cm)]} based on CT. The study demonstrated that the risk of resection failure reached 10% when the resection line was deeper than 30 mm from the lung surface. To overcome the problems arising from VAL-MAP 1.0, several new techniques have been developed.

## 6. ENB VAL-MAP

The steps of ENB VAL-MAP 1.0 are summarized in Figure 2. Lung map and marking locations are designed in the same way with VAL-MAP 1.0 using Synapse VincentVR (Fujifilm Medical). The locational information for each marking is then transferred to the ENB system (superDimension™ navigation system; Medtronic, Minneapolis, MN). A peripheral target for each marking is selected, and the pathway leading to the target is semiautomatically determined in the system. After this preparation, the patient is brought into the operation room and general anesthesia is induced with a single-lumen tracheal tube. A locatable guide-containing sensor and metal-tipped catheter are inserted into an extended working channel and advanced through the target under real-time navigation. Once the catheter reaches the target in the virtual image, the working channel is locked in place, and the catheter is removed. Thereafter, a metal-tip catheter preloaded with 0.7 mL of indigo carmine is inserted into the working channel under fluoroscopic guidance. The introduction of ENB to VAL-MAP has made it possible to perform preoperative mapping and surgery consecutively in an operating room during the same session of general anesthesia.

Moreover, on-site adjustment with ENB VAL-MAP eliminates the post-mapping CT scan requirement [36]. On-site adjustment is a procedure in which the information of the actual marking location obtained from the recorded video images is transferred to a portable workstation and replaced with the planned marking in the operating room within several minutes. This procedure has improved ENB VAL-MAP quality, achieving good outcomes comparable to those of VAL-MAP 1.0 (successful resection rate: 90.5%) [36]. However, the discrepancy between the planned and actual marking locations has not been measured objectively. Therefore, in the future, the accuracy of ENB VAL-MAP with on-site adjustment should be precisely evaluated by comparing it with the locational information of post-mapping CT.

## 7. VAL-MAP Using Indocyanine Green

Recently, we have shown that successful sublobar lung resection using VAL-MAP requires the creation of “three” intraoperative visible marks per established lung nodule [37]. Two hundred ten consecutive patients with two hundred fifty-six lesions who underwent VAL-MAP in our institution were retrospectively examined. This study showed that the successful resection rate was over 97% when three or more VAL-MAP marks could be identified during operation, while that was 79% when zero marks or one mark was visible during the surgery. This study also discussed that “four” VAL-MAP marks per nodule should be designed and established. Considering that approximately 90% of all the attempted VAL-MAP markings can be visible during the operation [11,27], the probability rate of identifying all the three marks, which are established using VAL-MAP, would be 73%. If four marks per lesion are placed, the probability rate of detecting three marks intraoperatively can increase to 95% [37].

In contrast, an increase in the number of VAL-MAP marks may cause bronchoscopic complications more often, leading to patient burden. Thus, strategies that can improve the success rate of marking detection during the surgery are indispensable. One of such strategies is the use of indocyanine green (ICG). ICG is a widely used fluorescent dye that is visualized with a near-infrared (NIR) thoracoscope (Figure 3). The vivid green color of ICG can be easily detected through the NIR scope during surgery, improving the success rate of VAL-MAP markings. Yanagiya et al. reported the usefulness of VAL-MAP using both ICG and indigo carmine (VAL-MAP dual staining) in five cases [38]. VAL-MAP dual staining is performed using 0.1 mL (0.25 mg) of Diagnogreen (Daiichi-Sankyo, Tokyo, Japan) and 1 mL (4 mg) of indigo carmine (Daiichi-Sankyo) for each marking. The small amount of ICG is enough for injection because of its high sensitivity with the NIR thoracoscope. We performed 72 VAL-MAP markings for 27 lesions in 20 patients using this method and demonstrated that VAL-MAP dual staining has improved the success rate of marking detection during surgery compared with VAL-MAP using indigo carmine alone (95.7% vs. 85.5%) without causing additional complications [39]. Another group performed VAL-MAP using ICG and CT contrast agents instead of indigo carmine (ICG VAL-MAP) [40]. This group used 0.1 mg Diagnogreen (Daiichi-Sankyo) and 66.6 mg Iopamiron (Bayer AG, Leverkusen, Germany) for each marking, and showed that 141 out of 142 attempted markings (99.3%) using ICG VAL-MAP were identifiable during surgery. The rate of marking detection was significantly higher than that of VAL-MAP 1.0 (525 out of 567 attempted markings; 92.6%).

Although VAL-MAP using ICG is likely to be quite useful and safe in increasing the detectability of markings, the use of ICG in VAL-MAP has certain limitations. One is that VAL-MAP using ICG is available only in institutions where an NIR scope is installed. Moreover, ICG is not applicable in patients with iodine hypersensitivity.

## 8. VAL-MAP Combining Microcoils (VAL-MAP 2.0)

The required resection margin was an independent risk factor for the successful resection of the targeted nodules in a multicenter prospective study [29]. Since VAL-MAP 1.0 is a two-dimensional marking technique with information limited to the pleural surface, acquisition of sufficient resection margins, especially for deeply located tumors, was a major concern. To overcome this issue, VAL-MAP 2.0—a three-dimensional mapping technique that involves intrabronchial placement of a microcoil—has been developed [41]. This technique combines bronchoscopic multispot dye-marking on the lung surface and microcoil placement in the peripheral bronchus, allowing three-dimensional mapping and sufficient resection margins in sublobar lung resection (Figure 4) [42]. VAL-MAP 2.0 is designed using a three-dimensional reconstruction of CT data and virtual bronchoscopic images in the same way as VAL-MAP 1.0. After the markings on the lung surface are established, the microcoil is placed under fluoroscopic guidance in a bronchus that is central to the target nodule for wedge resection and close to an intersegmental plane for segmentectomy. Thereafter, a post-mapping CT scan is performed, and three-dimensional images of actual marks, microcoil, and tumors are further reconstructed. The operation is conducted using three-dimensional images and fluoroscopy, which are used to visualize the microcoil and guide deep resection lines at stapling. A recent multicenter prospective single-arm study showed that successful resection was achieved in 64 of 65 resections (98.5%) with no severe adverse events associated with the VAL-MAP 2.0 procedure, although three microcoils showed major displacement after bronchoscopic placement among seventy five microcoils [41]. The limitations of this technique are that a fluoroscope must be used several times during surgery and microcoils are costly. Recently, a novel bronchoscopy-guided marking, similar to VAL-MAP 2.0, has been reported [42]. This study showed that four patients underwent bronchoscopic marking with a fiducial coil saturated with ICG using robot-assisted navigation bronchoscopy, and that all the lesions could be successfully resected without complications. This approach may be advantageous in that a fluoroscope is not required during the operation, but there is still concern that ICG in deeply located coils cannot be detected.

## 9. Summary of VAL-MAP

Table 2 summarizes the advantages and disadvantages of each VAL-MAP technique. Efficacy and good outcomes of VAL-MAP 1.0 have been demonstrated over the last 10 years. However, this technique has some disadvantages, including a prerequisite of mandatory post-mapping CT scans, occasional inability to see dye marks during surgery, and infrequent unsuccessful resection failure due to deep resection margins. Several new techniques have been developed to address these issues. ENB VAL-MAP with on-site adjustment can eliminate the need for a post-mapping CT scan althoughthere may be discrepancy between the planned and actual marking location. The use of ICG in VAL-MAP is not applicable for patients with iodine hypersensitivity but can augment the success rate of marking detection during surgery. VAL-MAP 2.0, with microcoil placement, can improve the accuracy of sublobar resection, particularly for deeply located tumors, though a fluoroscope is needed several times during surgery, and microcoils are costly. These new VAL-MAP techniques have many advantages that outweigh the disadvantages.

## 10. Conclusions

VAL-MAP has demonstrated efficacy and good outcomes over the last 10 years, and several new techniques have been developed to address the problems associated with conventional VAL-MAP. They are promising and beneficial for sublobar lung resection.

## Figures and Tables

**Figure 1 cancers-15-01971-f001:**
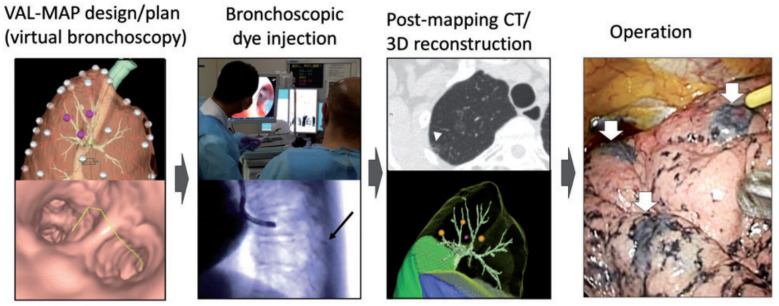
Steps of conventional virtual-assisted lung mapping [11].

**Figure 2 cancers-15-01971-f002:**
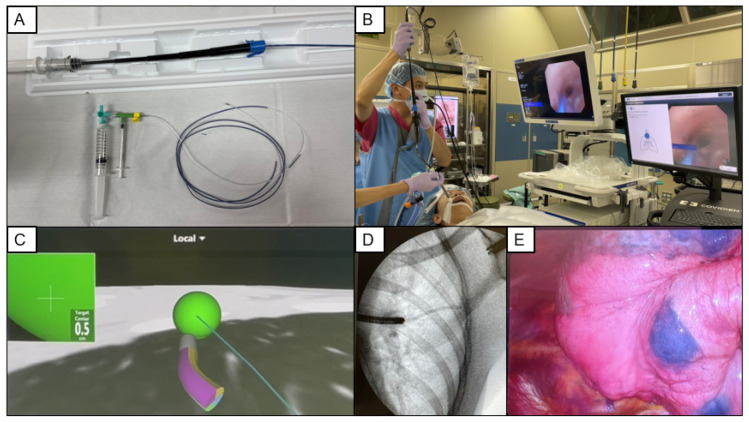
Steps of electromagnetic navigation bronchoscopy virtual-assisted lung mapping. (**A**) Locatable guide (LG)-containing sensor and metal-tipped catheter are used for electromagnetic navigation bronchoscopy (ENB) lung mapping. (**B**) The LG is inserted in an extended working channel (EWC) and advanced through the target under real-time navigation. (**C**) Once the LG/EWC reaches the target in the virtual image, the EWC is locked in place, and the LG is removed. (**D**) A metal-tip catheter preloaded with 0.7 mL of indigo carmine is inserted into the EWC under fluoroscopic guidance. (**E**) The introduction of ENB to virtual-assisted lung mapping makes it possible to perform the preoperative mapping and surgery consecutively in an operating room during the same session of general anesthesia.

**Figure 3 cancers-15-01971-f003:**
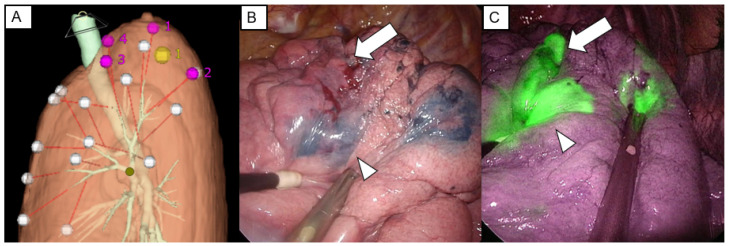
Virtual-assisted lung mapping using both indigo carmine and indocyanine green. (**A**) Four markings (pink circle) surrounding the nodule (yellow circle) are planned before virtual-assisted lung mapping using three-dimensional reconstruction of computed tomography data and virtual bronchoscopic images. (**B**) One marking with indigo carmine (arrowhead) is barely identifiable, and another marking (arrow) cannot be detected. (**C**) Indocyanine green can be easily identified using near-infrared thoracoscope.

**Figure 4 cancers-15-01971-f004:**
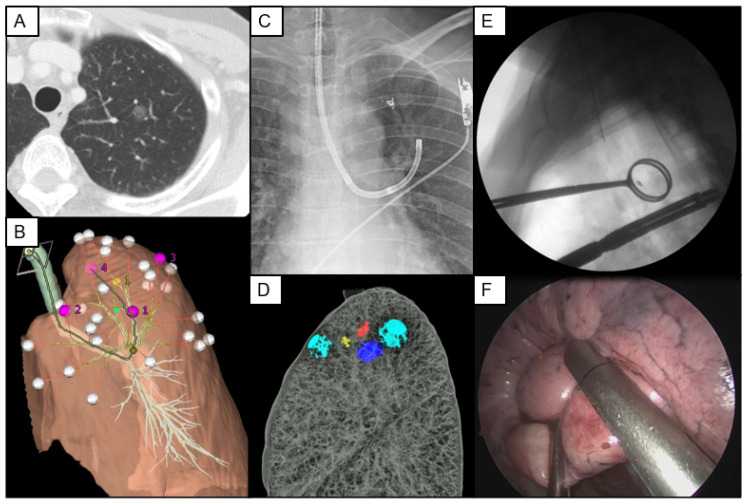
Steps of virtual-assisted lung mapping 2.0. (**A**) Virtual-assisted lung mapping (VAL-MAP) 2.0 is designed to be used for targets which are barely palpable and require a deep resection margin. (**B**) VAL-MAP 2.0 is designed using three-dimensional reconstruction of computed tomography (CT) data and virtual bronchoscopic images in the same way as VAL-MAP 1.0. (**C**) Microcoil is placed under fluoroscopic guidance in a bronchus which is central to the target nodule for wedge resection and close to an intersegmental plane for segmentectomy. (**D**) After a post-mapping CT scan is performed, three-dimensional images of actual marks (blue), microcoil (yellow), and tumor (red) are further reconstructed. (**E**) A fluoroscope is used to visualize the microcoil and guide deep resection lines at stapling during surgery. (**F**) The operation is conducted using three-dimensional images and fluoroscopy for guidance. This figure is reprinted from the reference [43].

**Table 1 cancers-15-01971-t001:** Advantages and disadvantages of CT-guided localization and bronchoscopic marking.

		Advantages	Disadvantages
**CT-guided localization**	Hook-wire [12,13,14,15]	λ Evidence from many studies λ No radiation exposure during surgery	λ Dislodging of the wire λ Discomfort during the waiting time to surgery
	Microcoils and fiducial markers [16,17,18,19]	λ No wire protruding extracorporeally after localization λ Flexibility of operating schedules	λ Fluoroscopic guidance during surgery λ High cost
	Dye marking [20,21,22]	λ Available without additional costs λ No radiation exposure	λ Needing immediate surgery after localization λ Anaphylaxis to dye
	Radiotracers [23,24]	λ Stability of materials for 24 h	λ Limitation of available facilities λ Possible radiation exposure
**Bronchoscopic marking**	Dye marking under ENB [25,26,27]	λ High success rate of marking λ Low rates of complications	λ Difficulty of localizing tumors when the marking fails λ High cost
	VAL-MAP 1.0 [10,25,28]	λ Making multiple markings easily λ Low rates of complications λ Low cost λ Efficiency in intraoperative adhesion cases	λ Post-mapping CT mandatory λ Invisible marks during surgery (10%) λ Unsuccessful resection for deeply located tumors

ENB, electromagnetic navigation bronchoscopy; VAL-MAP, virtual-assisted lung mapping.

**Table 2 cancers-15-01971-t002:** Advantages and disadvantages of each virtual-assisted lung mapping technique.

	Advantages	Disadvantages
**Conventional VAL-MAP** **(VAL-MAP 1.0)**	λ Good outcomes in two prospective multicenter studies [11,29] λ Evidence of long-term results [30]	λ Post-mapping CT mandatory [11] λ Invisible marks during surgery (10%) [29,32,37] λ Unsuccessful resection for deeply located tumors [29]
**ENB VAL-MAP** [36]	λ Elimination of post-mapping CT	λ Discrepancy between the planned and actual marking location [31]
**Use of ICG in VAL-MAP** [38,39,40]	λ Good success rate of marking detection	λ Not applicable for patients with iodine hypersensitivity λ Necessity of NIR scope
**VAL-MAP 2.0** **with microcoil replacement** [41]	λ Three-dimensional mapping λ Successful resection for deeply located tumor	λ Necessity of fluoroscope during surgery λ Cost of microcoils

CT, computed tomography; ENB, electromagnetic navigation bronchoscopy; ICG, indocyanine green; NIR, near-infrared; VAL-MAP, virtual-assisted lung mapping.

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
