# Peer review of "Ten-Year Outcome and Development of Virtual-Assisted Lung Mapping in Thoracic Surgery"

_cancers, 2023, doi:10.3390/cancers15071971_

Round 1

Reviewer 1 Report

Kudos to the authors for the work. The radiological technological development in the detection of small lung lesions is really impressive. Virtual-assisted lung mapping (VAL-MAP) combined with new CT (electromagnetic navigation bronchoscopy, (ICG) Indocianyne or microcoil mapping techniques proves to have many more advantages than disadvantages. The main difficulty is certainly the costs and availability of specific facilities and professionalism as well as hybrid rooms. Surely the use of these technical combinations help in the execution of sublobar resections and primarily in the identification of those parenchymal alterations not possible to palpate with minimally invasive technique

Author Response

Thanks for your great comments. We hope that this bronchoscopy-guided localization technique would help thoracic surgeons to perform sublobar lung resection more easily and safely. We added some information especially about CT-guided localization and some references on the other reviewer’s advice.

Reviewer 2 Report

The review paper in front of me is dedicated to a very important topic- Virtual-Assisted Lung  Mapping in Thoracic Surgery. However, this paper does not look like a review paper. First of all, only 30 references, and in half of them one of the authors is M. Sato.  Only four figures and only one table are not enough to demonstrate the Ten-Year Outcome and Development of Virtual-Assisted Lung  Mapping in Thoracic Surgery. The captions of Figures are also not appropriate.  In my opinion, this review article should be reviewed again after significant improvement.  

Author Response

The review paper in front of me is dedicated to a very important topic- Virtual-Assisted Lung Mapping in Thoracic Surgery. However, this paper does not look like a review paper. First of all, only 30 references, and in half of them one of the authors is M. Sato. 

>Thanks for your comment. We have added some information especially about CT-guided localization (Page 3 Line 113-Page 5 Line 175), a new table (Table 1), and some references (#13-24, and #42) on your advice although we are sure that this paper meets the criteria of review articles in Cancers (at least 30 references, two figures or tables). VAL-MAP is a relatively new localization technique for small pulmonary nodules, and many papers have been reported from Japan. That is why half of the related articles include M. Sato as a co-author, who originally developed this unique technique. However, some papers about VAL-MAP have been recently reported from other countries, and we hope that this bronchoscopy-guided localization technique would gain widespread use in the following decade.

Only four figures and only one table are not enough to demonstrate the Ten-Year Outcome and Development of Virtual-Assisted Lung Mapping in Thoracic Surgery. The captions of Figures are also not appropriate.  In my opinion, this review article should be reviewed again after significant improvement. 

>Thank you for your suggestion. We have added a table (Table 1) to explain the advantages and disadvantage of different techniques including VAL-MAP. We believe this comparison with other techniques would be beneficial to readers’ understanding of this topic as a review article. Namely, the comparison with other techniques (Table 1) in addition to comparison within VAL-MAP (Table 2; original Table 1) highlights the impact of the 10-year development of VAL-MAP. We believe the four figures represent all the important progress of VAL-MAP in 10 years and, once again, description of the techniques in the context of other localization techniques would provide comprehensive explanation about all the up-to-date techniques of VAL-MAP in this paper.

Round 2

Reviewer 2 Report

Publish in the present form.